# Estimating the disease burden of Korean type 2 diabetes mellitus patients considering its complications

**Juyoung Kim**[1,2], **Seok-Jun Yoon**[3], **Min-Woo Jo**[1,2]*

**1** Asan Medical Institute of Convergence Science and Technology, Asan Medical Center, University of Ulsan College of Medicine, Seoul, South Korea, **2** Department of Preventive Medicine, University of Ulsan College of Medicine, Seoul, South Korea, **3** Department of Preventive Medicine, Korea University College of Medicine, Seoul, South Korea

* mdjominwoo@gmail.com

## Abstract

### Background

The burden of diabetes is considerable not only globally but also nationally within Korea. The Global Burden of Disease study derived the disability-adjusted life years (DALYs) of diabetes depending on its complications as individual severity using prevalence-based approach from 2017. Conversely, the Korean National Burden of Disease study based on an incidence-based approach does not incorporate the severity of diseases. This study aimed to simulate incidence-based DALYs of type 2 diabetes mellitus (T2DM), given diabetic complications as disease severity using a Markov model.

### Methods

We developed a model with six Markov states, including incident and existing prevalent cases of diabetes and its complications and death. We assumed that diabetes and its complications would not be cured. The cycle length was one year, and the endpoint of the simulation was 100 years. A 5% discount rate was adopted in the analysis. Transition cases were counted by 5-year age groups above 30 years of age. Age- and sex-specific transition probabilities were calculated based on the incident rate.

### Results

The total DALY estimates of T2DM were 5,417 and 3,934 per 100,000 population in men and women, respectively. The years of life lost in men were relatively higher than those in women in most age groups except the 80–84 age group. The distribution of years lived with disability by gender and age group showed a bell shape, peaking in the 55–59 age group in men and 65–69 age group in women.

### Conclusions

The burden of T2DM considering its complications was larger compared to the outcomes from previous studies, with more precise morbid duration using the Markov model.

**Data Availability Statement:** The data underlying the results presented in the study are available from the National Health Insurance Sharing Service (https://nhiss.nhis.or.kr/bd/ay/bdaya001iv.do).

**Funding:** This study was supported by a grant from the Korean Health Technology R&D Project (HI18C0446). The funders had no role in study design, data collection and analysis, decision to publish, or preparation of the manuscript.

**Competing interests:** The authors have declared that no competing interests exist.

## Introduction

Diabetes is a leading cause of disability-adjusted life years (DALYs) accounting for 2.6% of global DALYs in 2017, according to the Global Burden of Disease (GBD) study [1]. The global burden of diabetes ranked twentieth in 1990 (534.35 DALYs per 100,000) and jumped to eighth in 2017 (867.81 DALYs per 100,000). In 2017, the years lived with disability (YLDs) of diabetes ranked as the fourth and ninth leading cause in men and women, respectively [2]. Considering the substantial prevalence of diabetes, the global diabetes burden is expected to be greater in the future [3, 4]. In the Korea National Burden of Disease (KNBD) study, DALYs and YLDs attributable to diabetes have been ranked the highest among the top 20 leading causes since 2007, and the burden of diabetes is expected to increase, given the increasing prevalence [5]. Type 2 diabetes mellitus (T2DM) is the most common type, accounting for over 90% of diabetes, and largely preventable [6]. Thus, we confined the scope of the disease burden of diabetes only to T2DM.

Diabetic complications cannot be ignored in the estimation of disease burden considering premature mortality due to macrovascular complications such as cardiovascular diseases [7]. The disease severity, such as uncomplicated and complicated diabetes, should be incorporated for more elaborate DALY estimation. The GBD 2017 incorporated diabetic complications such as vision loss and diabetic neuropathy in prevalence-based DALY estimation [2]. In contrast, the latest KNBD study obtained DALYs based on an incidence-based approach, which is a forward-looking approach [8, 9]. There has been no effort to calculate the burden of disease considering its severity within the KNBD study. In addition, the morbid duration derived from the DisMod II, a tool used to estimate the disease period in a previous GBD and KNBD studies, was underestimated [10, 11]. A multi-state Markov model, frequently used in decision analyses including economic analysis, can be used to calculate morbid duration including separate health states [12]. DALYs are measuring the burden of disease at global, national, and local level and also used in economic evaluations [13–15]. These allow us to investigate the national burden of diabetes adopting a Markov model. This study aimed to estimate DALYs of T2DM, given diabetic complications, in a Korean population based on an incidence-based approach using a multi-state Markov model.

## Materials and methods

### Construction of a Markov model

A Markov model was introduced to estimate the incidence-based DALYs of patients with T2DM considering complications such as retinopathy, neuropathy, ischemic heart disease, and so on (Fig 1). Considering the disability weights (DWs) of diabetes according to the severity of the disease in a previous study (DW of diabetes with complications = 0.663, DW of diabetes without complications = 0.334), our model is comprised of six Markov states representing the health states of diabetic patients: incident cases of diabetes, existing prevalent cases of diabetes, incident cases of complications, existing prevalent cases of complications, death due to diabetes, and death due to other causes [16]. The movements between these Markov states (i.e., transition) are presented in Fig 1. A patient in an initial state (i.e., incident cases of diabetes) has five possible transitions in the second cycle: the patient can (1) remain in this state, (2) progress to a state of incident cases of complications, (3) progress to death due to diabetes, (4) progress to the state of death due to diabetes, or (5) progress to the state of death due to other causes. The time interval of transition between states (i.e., cycle length) was one year. To reflect the opportunity cost, a 5% discount rate was adopted in the analysis. In our model, diabetes and diabetic complication states were assumed to be irreversible, considering the

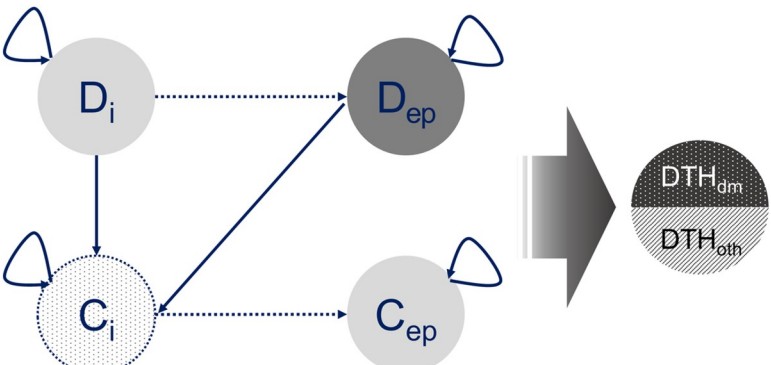

**Fig 1. Markov model of type 2 diabetes including complications.** Di, incident cases of type 2 diabetes; Dep, existing prevalent cases of type 2 diabetes, Ci, incident cases of complications, Cep, existing prevalent cases of complications; DTH dm, death due to diabetes; DTH oth, death due to other causes. Solid arrows: Transition probabilities directly derived from the NHIS data. Dotted arrows: Hypothetical transitions explaining the conversion of transition probabilities from incidence to existing prevalence after five cycles.

characteristics of diabetes as a chronic condition [17, 18]. Although new treatments for T2DM, such as bariatric surgery, were introduced, we did not consider them because they were neither general treatment procedures nor being covered by the National Health Insurance system in the index year of our study (i.e., 2016). Our model also assumed that incident cases are treated as existing prevalent cases after five cycles, considering evidence showing different impacts on the occurrence of complications and mortality 5 years after the diagnosis of diabetes [19–22]. Among the 16 conceptual transitions in Fig 1, the two transition paths (dotted lines) show that the incidence was used as the transition probability for the first five cycles, after which the prevalence was applied. No transitions occurred once a death state was entered (i.e., absorbing state). The endpoint of the simulation was 100 years.

## Data source and parameters

The National Health Insurance Service (NHIS) database, representing the whole Korean population, was used to derive most parameters, including diagnostic information and the causes of death. Raw data from the NHIS were used to derive the number of incident cases in 2016, the NHIS-National Sample Cohort (NHIS-NSC) was used to determine transition probabilities between 2009 and 2015 [23]. In addition to the NHIS data, we also utilized a complete life table from the Korean Statistical Information Service to find the general mortality by gender and age in 2009 and 2016 [24].

T2DM cases were defined as follows: diagnosed with the International Classification of Disease (ICD)-10 code of T2DM (ICD-10 code, E11–E14); prescribed anti-diabetic drugs (ingredient code listed in S1 Table) with reference to previous publications, and over 30 years old [25, 26]. We only included patients with T2DM aged over 30 years for the following reasons: (1) T2DM is common in adults, and (2) the incidence of T2DM under the age of 30 years only comprises around 0.1% in Korea [27–29]. Incident cases were defined as T2DM patients who did not use medical services due to T2DM in the last three years (washout period), considering the features of the claim data. We chose a 3-year washout period to extract true incident cases considering previous research [30]. We also introduced the concept of existing prevalent cases, defined as the complementary set of incident cases from the whole prevalent cases. We used the existing prevalent case group to clearly differentiate diabetic patients treated for a certain

period from the classical prevalent cases, in light of previous outcomes showing differences in the features of incident and prevalent cases [11, 31].

Complications were determined based on the ICD-10 codes, which were obtained via a literature review and consultation with an endocrinologist, as listed in S2 Table [32–36]. Listed diagnosis codes for diabetic complications were categorized into two groups: diabetes-related complications irrespective of age and diabetes-related complications that occurred over 60 years of age. To calculate YLLs in patients with T2DM, a case recorded as T2DM (E11–E14) on the death certificate was only regarded for our estimation.

An individual date of the first diagnosis in the initial year was regarded as the index date to calculate the length of state for each incident case, whereas the first day of the initial year was treated as the index date for existing prevalent cases. The duration was calculated in person-days and then converted to person-years. Transition events were counted in 5-year age groups (from 30–34 to 85 years or older) and by gender. The transition rate by gender and age group was calculated from the duration and number of cases and then converted into transition probabilities using the following formula [37]:

$$p = 1 - \exp(-rt)$$

$$p = probability; \ r = rate; \ t = time(duration, cycle)$$

For unstable parameters such as the transition probabilities to death, combined-year probabilities were used. First, we included five annual cohorts (from 2010 to 2015) consisting of complications from incident cases in 2009. This is because only a few initial cases with complications were found, and the transition events to death were also rare. Second, we used smoothing methods by merging estimates of neighbor age groups to calculate the aggregate probabilities. We intended to reduce the impact of outliers originating from small samples in a certain age group by replacing aggregate figures. In terms of mortality, death probabilities due to causes other than diabetes by gender and age group were compared with general mortality probabilities. If the mortality probability was less than that in the general populations of the same gender and age group, the general probability substituted the corresponding figures. The lower mortality of diabetic patients than that of the general population is also likely attributable to small populations in specific age groups. Therefore, zero mortalities in the 30–34 age group, regardless of the cause of death, were replaced with general death probabilities as well.

## Cohort simulation and sensitivity analyses

The YLL estimates were calculated as the number of deaths due to diabetes multiplied by the duration between the age at premature death and the standard life expectancy. For the YLD component, the disease duration of diabetes was calculated separately depending on the occurrence of complications, and the number of incident cases was multiplied by the relevant DW and the duration calculated. A range of values for YLDs and DALYs was obtained using a 95% confidence interval for DW.

Reverse transitions between the diagnosis of diabetes and complications in the incident cases of diabetes were handled as follows:

1. A group of diabetes-related complications irrespective of age (E codes) was incorporated into existing prevalent cases of diabetes. Complication codes beginning with E are diabetic complication codes, implying the occurrence of diabetes with complications. Therefore, we treated these cases as existing prevalent cases of diabetes.

2. When a patient with a history of stroke or ischemic heart disease is diagnosed with diabetes, it is uncertain whether these are diabetic complications or not. Therefore, a group of diabetes-related complications that occurred over 60 years of age (other than E codes) was subjected to a sensitivity analysis.

The disease duration between diabetes and its complications was treated as 0 day (min), 180 days (median), and 365 days (max). With respect to the minimum assumption on the disease duration (from the index date of diabetes to the index date of complications), we assumed that the complications were diagnosed on the same day as the diabetes. For the maximum assumption, the designated cycle (365 days) was used. Then, a comparison between DALYs depending on the level of disease duration was carried out. The parameter calculation was conducted using SAS Enterprise Guide version 7.1 (SAS Institute, Cary, NC, USA), and the cohort simulation was conducted using Microsoft 365 Excel. This research was approved by the Institutional Review Board of Korea University (2019-0182-01).

## Results

In 2016, the prevalence of diabetes (≥30 years old) was 10.7% (106.7 per 1,000 people, 1.8 million) and 8.4% (84.4 per 1,000 people, 1.5 million) in men and women, respectively. The incident cases were 0.021 million (12.5 per 1,000 people) in males and 0.015 million (8.7 per 1,000 people) in females. The incidence of diabetes between 2009 and 2015 was 1.3% (12.5 per 1,000 people) and 0.9% (8.7 per 1,000 people) in men and women, respectively. The age-specific incidence rate of males showed an overall increasing tendency with age and exceeded the aggregate incidence rate (12.5 per 1,000 people) in the 50–54 age group (Table 1). In females, the peak age group was 70–79 years (17.0 per 1,000), with a steady decline thereafter (Table 2).

The overall transition probabilities of complications, from either incident cases or existing prevalent cases, were higher in females. In both genders, the probabilities of transition to complications were mostly higher from the incident cases for diabetes compared to transitions from the existing prevalent cases. However, the transition probabilities to complications from the existing prevalent cases were higher than those from the incident cases in male patients aged over 75 years. For mortality probabilities (from incident cases of T2DM to death, from existing prevalent cases to death, from incident cases of complications to death, and from existing prevalent cases of complications to death), the following overall features were identified: an increasing tendency with age, higher probabilities in women than men, and lower probabilities of diabetes-related death compared to death due to other causes. Among the segmented mortalities, the transition probabilities from incident cases of diabetes to deaths other than diabetes were the highest in all age groups except 30–34 years in both men and women.

In 2016, the total DALY estimates of diabetic patients per 100,000 population in men and women were estimated at 5,417 and 3,934, respectively. The distribution of DALY estimates (per 100,000 population) by gender and age group demonstrated bell shapes peaking in the 55–59 age group in males (11,520 DALYs per 100,000 population) and in the 65–69 age group in females (9,795 DALYs per 100,000 population), as seen in Fig 2 (S1 Fig, total estimated DALYs). In the comparison between genders, men showed a relatively high disease burden due to T2DM from the early 30s to the early 60s compared to women, and the opposite trend was observed in people aged 65 and over.

Overall, the estimated YLLs per 100,000 in men were relatively higher than in women (Fig 3; S2 Fig, total estimated YLLs). This tendency was observed in the age- and sex-specific YLL estimates (per 100,000 population) in all age groups except for 80–84 years. The distribution of the YLL estimates (per 100,000 population) by gender and age group showed a bell shape, with the peak estimate in the 60–64 age group (2,430 YLLs per 100,000 population in males vs.

**Table 1. Transition probabilities of males by age group.**

| Age | IC | IR [a] | Transition probabilities | | | | | | | | | |
|---|---|---|---|---|---|---|---|---|---|---|---|---|
| | | | Di→Ci | Di→DTH dm | Di→DTH others | Dep→Ci | Dep→DTH dm | Dep→DTH others | Ci→DTH dm | Ci→DTH others | Cep→DTH dm | Cep→DTH others |
| 30–34 | 6,736 | 3.7 | 0.171 | 0.004 | 0.007 | 0.106 | 0.002 | 0.001 | 0.001 | 0.004 | 0.010 | 0.005 |
| 35–39 | 10,909 | 5.4 | 0.121 | 0.000 | 0.014 | 0.105 | 0.001 | 0.006 | 0.001 | 0.006 | 0.002 | 0.008 |
| 40–44 | 18,310 | 8.5 | 0.113 | 0.001 | 0.017 | 0.093 | 0.002 | 0.006 | 0.001 | 0.010 | 0.001 | 0.008 |
| 45–59 | 25,951 | 11.5 | 0.116 | 0.001 | 0.017 | 0.098 | 0.001 | 0.008 | 0.001 | 0.010 | 0.002 | 0.009 |
| 50–54 | 30,739 | 14.5 | 0.113 | 0.002 | 0.025 | 0.102 | 0.001 | 0.012 | 0.001 | 0.010 | 0.003 | 0.011 |
| 55–59 | 34,156 | 16.8 | 0.117 | 0.001 | 0.027 | 0.103 | 0.002 | 0.017 | 0.002 | 0.010 | 0.004 | 0.014 |
| 60–64 | 27,605 | 19.0 | 0.189 | 0.003 | 0.047 | 0.152 | 0.002 | 0.023 | 0.001 | 0.015 | 0.004 | 0.017 |
| 65–69 | 20,110 | 19.2 | 0.175 | 0.003 | 0.070 | 0.150 | 0.003 | 0.043 | 0.009 | 0.024 | 0.007 | 0.031 |
| 70–74 | 15,486 | 19.5 | 0.140 | 0.003 | 0.115 | 0.139 | 0.009 | 0.058 | 0.009 | 0.049 | 0.012 | 0.050 |
| 75–79 | 11,132 | 19.4 | 0.127 | 0.002 | 0.160 | 0.131 | 0.010 | 0.088 | 0.003 | 0.059 | 0.015 | 0.067 |
| 80–84 | 5,988 | 20.0 | 0.078 | 0.010 | 0.240 | 0.090 | 0.032 | 0.129 | 0.008 | 0.108 | 0.024 | 0.117 |
| 85+ | 2,868 | 20.4 | 0.043 | 0.015 | 0.265 | 0.059 | 0.033 | 0.144 | 0.028 | 0.205 | 0.037 | 0.122 |

IC, incident cases; IR, incident rate; Di, incident cases of type 2 diabetes; Dep, existing prevalent cases of type 2 diabetes; Ci, incident cases of complications; Cep, existing prevalent cases of complications; DTH, death.

[a] per 1,000 people.

1,724 YLLs per 100,000 population in females). The distribution of YLD estimates by gender and age group was also bell shaped, peaking in the 55–59 age group in men (9,201 YLDs per 100,000 population) and 65–69 age group in women (8,173 DALYs per 100,000 population). The decomposition of YLD estimates per 100,000 populations according to complications is displayed in Fig 4 (S3 Fig, total estimated YLDs). Subtle differences in the YLD estimates without complications among age groups were a common feature in both sexes. Conversely, distinct bell shapes were observed in YLDs with complications, climbing in the 60–64 age group in males (6,718 YLDs per 100,000 population) and 65–69 age group in females (6,411 YLDs per 100,000 population).

In the sensitivity analysis regarding disease duration after the initial occurrence of diabetes and complications, noticeable changes were not observed in males or females (Tables 3 and 4). Overall, the difference was less than 10 DALYs per 100,000 population in comparison with the initial disease duration (minimum assumption).

## Discussion

In this study, the DALYs of T2DM patients were estimated to reflect complications using the Markov model. The DALYs were 5,417 and 3,934 per 100,000 in men and women, respectively. Our DALY estimates were more than five times the global estimates [1, 38]; they were also greater than those in the previous KNBD study [8]. Nevertheless, the patterns of DALYs,

**Table 2. Transition probabilities of females by age group.**

| Age | IC | IR [a] | Transition probabilities | | | | | | | | | |
|---|---|---|---|---|---|---|---|---|---|---|---|---|
| | | | Di→Ci | Di→DTH dm | Di→DTH others | Dep→Ci | Dep→DTH dm | Dep→DTH others | Ci→DTH dm | Ci→DTH others | Cep→DTH dm | Cep→DTH others |
| 30–34 | 3,940 | 2.2 | 0.157 | 0.005 | 0.011 | 0.123 | 0.004 | 0.003 | 0.001 | 0.001 | 0.013 | 0.007 |
| 35–39 | 4,671 | 2.4 | 0.134 | 0.004 | 0.039 | 0.121 | 0.000 | 0.003 | 0.001 | 0.001 | 0.003 | 0.018 |
| 40–44 | 6,990 | 3.3 | 0.150 | 0.002 | 0.039 | 0.117 | 0.000 | 0.003 | 0.001 | 0.002 | 0.003 | 0.018 |
| 45–59 | 11,690 | 5.3 | 0.164 | 0.002 | 0.039 | 0.124 | 0.000 | 0.005 | 0.001 | 0.003 | 0.005 | 0.018 |
| 50–54 | 17,612 | 8.6 | 0.130 | 0.003 | 0.040 | 0.121 | 0.000 | 0.006 | 0.001 | 0.005 | 0.004 | 0.018 |
| 55–59 | 22,627 | 11.1 | 0.143 | 0.002 | 0.040 | 0.127 | 0.000 | 0.008 | 0.001 | 0.005 | 0.005 | 0.018 |
| 60–64 | 21,202 | 14.0 | 0.185 | 0.004 | 0.055 | 0.175 | 0.002 | 0.010 | 0.001 | 0.005 | 0.005 | 0.020 |
| 65–69 | 17,810 | 15.7 | 0.201 | 0.002 | 0.064 | 0.170 | 0.002 | 0.016 | 0.002 | 0.014 | 0.006 | 0.027 |
| 70–74 | 16,503 | 17.0 | 0.196 | 0.002 | 0.092 | 0.170 | 0.005 | 0.031 | 0.002 | 0.019 | 0.008 | 0.032 |
| 75–79 | 14,176 | 17.0 | 0.152 | 0.001 | 0.092 | 0.154 | 0.008 | 0.052 | 0.006 | 0.047 | 0.007 | 0.035 |
| 80–84 | 9,409 | 16.6 | 0.136 | 0.006 | 0.141 | 0.113 | 0.015 | 0.093 | 0.003 | 0.073 | 0.008 | 0.042 |
| 85+ | 6,245 | 15.0 | 0.094 | 0.007 | 0.141 | 0.058 | 0.037 | 0.102 | 0.016 | 0.109 | 0.015 | 0.050 |

IC, incident cases; IR, incident rate; Di, incident cases of type 2 diabetes; Dep, existing prevalent cases of type 2 diabetes; Ci, incident cases of complications; Cep, existing prevalent cases of complications; DTH, death.

[a] per 1,000 people.

YLLs, and YLDs by gender and age were similar to those in previous studies. These differences were attributable to the efforts to calculate the burden of diabetes more precisely, by including diabetic complications as a separate substate of T2DM with noncomplicated diabetes [17]. This attempt was based on the greater impact on mortality from complications in diabetic patients, rather than diabetes itself [39].

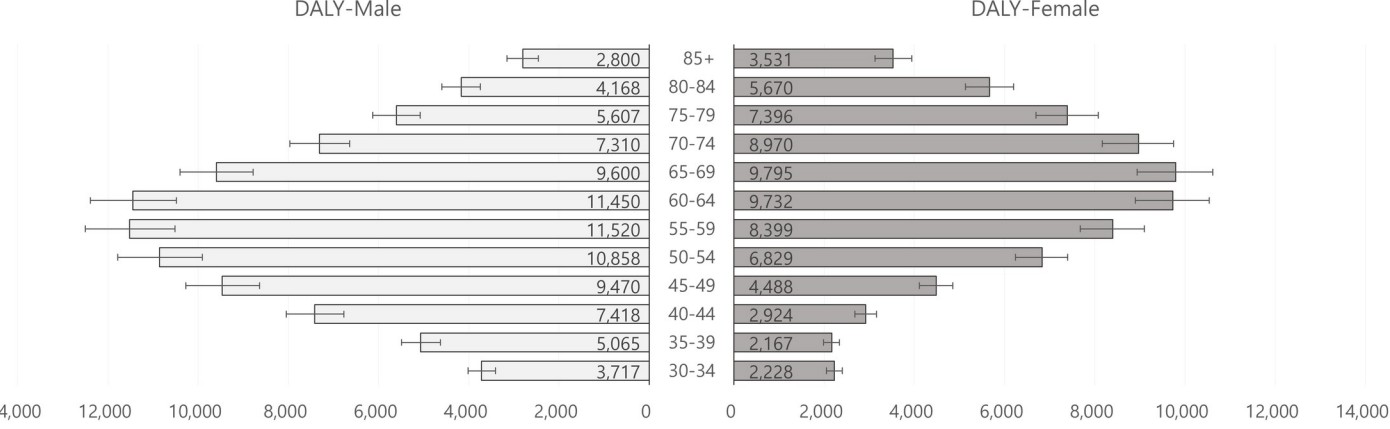

**Fig 2. Estimated DALYs per 100,000 population by gender and age group.** DALYs, disability-adjusted life years. Unit: DALYs per 100,000 population.

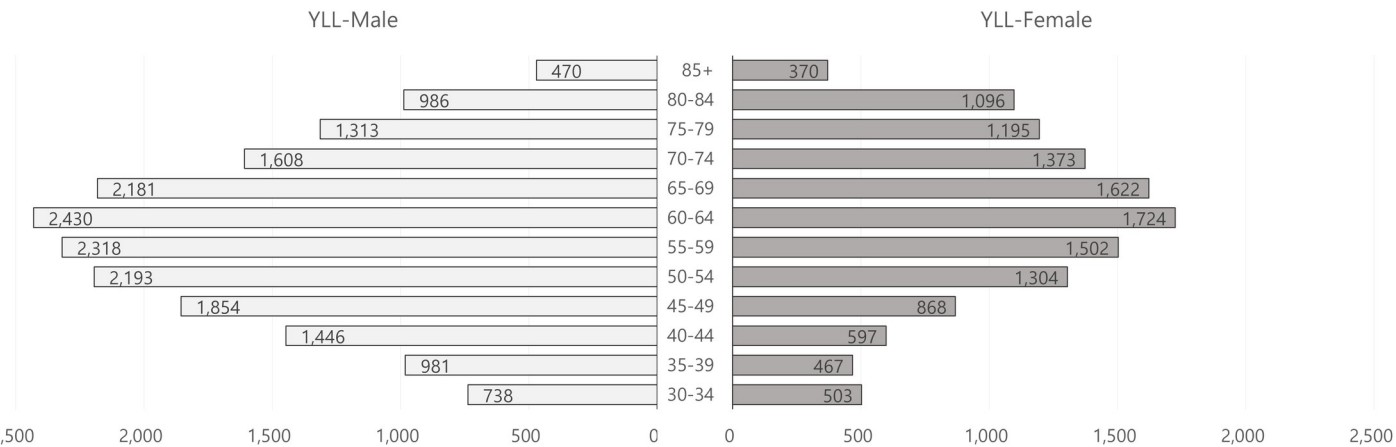

**Fig 3. Estimated YLLs per 100,000 population by gender and age group.** YLLs, years of life lost. Unit: YLLs per 100,000 population.

A Markov model was used to estimate the DALYs of diabetic patients considering related complications, which had a few methodological strengths, as a different approach was adopted to estimate DALYs in T2DM patients. First, we could construct a natural history model for the disease, including its complications with the Markov model, whereas DisMod II can only calculate the morbid duration of a single disease. We only considered one health state with diabetic complications, but in our next study we will be able to reflect various complicated states if a DW for each health state is available. Second, we obtained the YLDs and YLLs from a single Markov model that reflects our cohort of patients with diabetes. In contrast, the GBD study obtained the YLDs and YLLs from different sources in the calculation of the DALYs. Such discordance in the data source may lead to substantial errors—for example, a person who died from T2DM may not have been officially diagnosed with T2DM while alive.

There were some issues with the parameters. First, we defined existing prevalent cases in a way that separated incident and prevalent cases. This was an attempt to reflect the potential difference in survival rates between incident cases and existing prevalent cases based on previous research [11]. Second, the concept of person-years was used to alleviate the problem caused by a cycle in the Markov model that derived transition probabilities [40, 41]. For example, we calculated days (578 days) between the date of death (December 31, 2015) and the

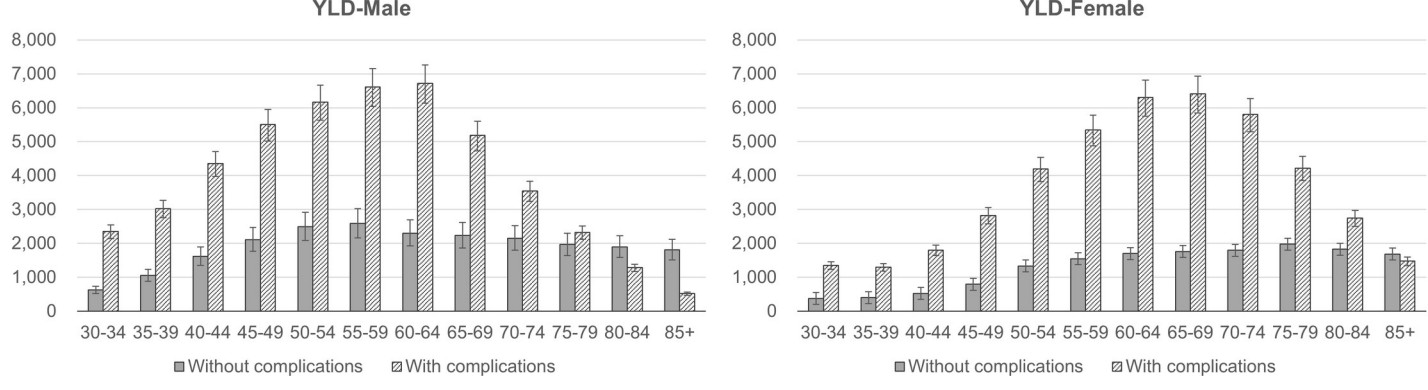

**Fig 4. Estimated YLDs per 100,000 population depending on complications.** YLDs, years lived with disability. Unit: YLDs per 100,000 population.

**Table 3. Results of sensitivity analysis in males.**

| Age | Sensitivity analysis (min) | | | Sensitivity analysis (median) | | | Sensitivity analysis (max) | | |
|---|---|---|---|---|---|---|---|---|---|
| | LB | DALY | UB | LB | DALY | UB | LB | DALY | UB |
| 30–34 | 3407.7 | 3716.8 | 4015.4 | 3407.7 | 3716.8 | 4015.4 | 3407.7 | 3716.8 | 4015.4 |
| 35–39 | 4626.0 | 5065.0 | 5492.0 | 4626.0 | 5065.0 | 5492.0 | 4626.0 | 5065.0 | 5492.0 |
| 40–44 | 6770.2 | 7417.7 | 8048.7 | 6770.2 | 7417.7 | 8048.7 | 6770.2 | 7417.7 | 8048.7 |
| 45–49 | 8640.3 | 9469.6 | 10278.4 | 8640.3 | 9469.6 | 10278.4 | 8640.3 | 9469.6 | 10278.4 |
| 50–54 | 9907.1 | 10857.8 | 11786.2 | 9907.1 | 10857.8 | 11786.2 | 9907.1 | 10857.8 | 11786.2 |
| 55–59 | 10515.2 | 11519.7 | 12499.8 | 10515.2 | 11519.7 | 12499.8 | 10515.2 | 11519.7 | 12499.8 |
| 60–64 | 10483.3 | 11450.1 | 12390.2 | 10482.9 | 11449.8 | 12390.0 | 10482.5 | 11449.5 | 12389.8 |
| 65–69 | 8778.0 | 9599.6 | 10403.3 | 8778.1 | 9600.1 | 10404.2 | 8778.3 | 9600.6 | 10405.1 |
| 70–74 | 6645.1 | 7310.3 | 7967.1 | 6646.0 | 7311.5 | 7968.6 | 6646.9 | 7312.7 | 7970.1 |
| 75–79 | 5079.7 | 5607.3 | 6132.7 | 5081.7 | 5609.6 | 6135.3 | 5083.7 | 5612.0 | 6138.0 |
| 80–84 | 3743.4 | 4168.3 | 4596.8 | 3743.4 | 4168.3 | 4596.8 | 3743.4 | 4168.3 | 4596.8 |
| 85+ | 2456.4 | 2799.9 | 3151.2 | 2456.4 | 2799.9 | 3151.2 | 2456.4 | 2799.9 | 3151.2 |

Min, minimum; Max, maximum; LB, lower bound; UB, upper bound; DALY, disability-adjusted life year.

index for complication occurrence (June 1, 2014). Thus, we could obtain more accurate probabilities close to the real duration. Lastly, regarding the incident probabilities of complications, we aggregated the cohort data of complicated incident cases from 2010 to 2014 and then calculated cumulative transition probabilities with summed person-years by gender and age group. This was because the proper probabilities were not available using a single-year cohort due to the low occurrence of complications for specific sex and age groups. We used a practical measure to achieve measurable and meaningful transition probabilities from the obtainable data.

The DALY estimates of male and female T2DM patients in 2016 were almost twice the incidence-based DALY estimates using DisMod II in 2015, with 0.593 DW for diabetes (males, 2,841 DALYs per 100,000 population; females, 2,048) [8, 42]. When compared to DALYs using DisMod II in 2016, a difference of 2.6 times was observed in our most recent analysis. In

**Table 4. Results of sensitivity analysis in females.**

| Age | Sensitivity analysis (min) | | | Sensitivity analysis (median) | | | Sensitivity analysis (max) | | |
|---|---|---|---|---|---|---|---|---|---|
| | LB | DALY | UB | LB | DALY | UB | LB | DALY | UB |
| 30–34 | 2048.0 | 2228.0 | 2402.1 | 2048.0 | 2228.0 | 2402.1 | 2048.0 | 2228.0 | 2402.1 |
| 35–39 | 1987.5 | 2167.2 | 2341.5 | 1987.5 | 2167.2 | 2341.5 | 1987.5 | 2167.2 | 2341.5 |
| 40–44 | 2680.0 | 2924.1 | 3160.4 | 2680.0 | 2924.1 | 3160.4 | 2680.0 | 2924.1 | 3160.4 |
| 45–49 | 4109.8 | 4487.9 | 4853.7 | 4109.8 | 4487.9 | 4853.7 | 4109.8 | 4487.9 | 4853.7 |
| 50–54 | 6243.0 | 6829.3 | 7398.2 | 6243.0 | 6829.3 | 7398.2 | 6243.0 | 6829.3 | 7398.2 |
| 55–59 | 7676.0 | 8399.0 | 9099.0 | 7676.0 | 8399.0 | 9099.0 | 7676.0 | 8399.0 | 9099.0 |
| 60–64 | 8900.0 | 9732.0 | 10536.1 | 8900.2 | 9732.3 | 10536.6 | 8900.4 | 9732.6 | 10537.0 |
| 65–69 | 8944.3 | 9795.3 | 10618.1 | 8944.3 | 9795.4 | 10618.4 | 8944.4 | 9795.6 | 10618.6 |
| 70–74 | 8166.2 | 8969.5 | 9748.6 | 8168.0 | 8971.6 | 9751.0 | 8169.7 | 8973.7 | 9753.4 |
| 75–79 | 6700.3 | 7395.5 | 8077.1 | 6699.6 | 7395.0 | 8076.8 | 6698.9 | 7394.4 | 8076.5 |
| 80–84 | 5128.6 | 5670.0 | 6205.8 | 5130.1 | 5672.0 | 6208.4 | 5131.7 | 5674.1 | 6211.0 |
| 85+ | 3124.3 | 3531.0 | 3938.9 | 3127.8 | 3535.3 | 3944.0 | 3131.4 | 3539.7 | 3949.3 |

Min, minimum; Max, maximum; LB, lower bound; UB, upper bound; DALY, disability-adjusted life year.

addition, when compared to global estimates of T2DM, our DALY estimates were five times greater [1, 38]. The outcome of the greater disease burden complied with our expectations, in light of the longer disease duration Cho et al. found using a Markov model [11]. In addition, our new approach of introducing diabetes-related complications also, to some extent, had an impact on the increased DALYs. For a more accurate estimation, we used separate DWs for diabetes with complications (0.663) and without complications (0.334) to reflect the impact of complications in diabetic patients, which we ascertained from the literature [43, 44]. In terms of the existing prevalent cases, it was difficult to designate logically appropriate index dates. Therefore, the first day of the year was regarded as the individual index data of existing prevalent cases of diabetes and complications, despite the fact that the disease would have initiated far earlier. This assumption regarding existing prevalent cases could have made the transition probabilities greater than the real probabilities, ultimately resulting in increased estimates of DALYs. In contrast, only mortalities due to diabetes identified from cause-of-death data were counted when calculating the YLLs. The DALY estimates could be more accurate and logical measures considering that the YLLs cover only deaths due to diabetes based on cases of death, preventing overestimation of the YLLs.

Some limitations of this study are as follows. First, diabetic complications were aggregated to calculate DALYs in patients with T2DM because we had only one DW for diabetic complications. However, there could be some differences in DWs among diabetic complications, such as retinopathy, neuropathy, and other macrovascular complications, considered in this study. Therefore, there is a need to obtain DWs on these states to ensure more valid DALYs for diabetic patients. Second, there might have been biases due to the NHIS-NSC data structure. Because the database included only information on the death month but gave the date of medical utilization, the morbid durations could be biased. Therefore, we conducted sensitivity analyses on this issue and identified that the biases were not large and could be accepted. In addition, if the exact death date can be obtained, the biases can be reduced. Third, data from the existing prevalent group could have created some biases. To construct a natural history model of T2DM, we developed a Markov model, which required transitional probabilities from prevalent non-complicated T2DM to complicated T2DM or the death state. The morbid histories of the existing prevalent groups were dependent on their incident ages. For example, for the early-30s incident groups, patients in existing prevalent states in their early 50s had already experienced T2DM for 20 years. However, in this study, all existing prevalent groups were diabetic patients with more than three years of morbid duration; therefore, their mortality or complication incidences could have been relatively lower than the actual mortality or incidence, which might have reduced the total DALYs of T2DM patients.

## Conclusions

This study estimated the DALYs of T2DM patients in Korea using a Markov model considering T2DM complications. As a result, the estimated burden of disease in patients with T2DM was larger than that in previous studies because this study included its complications. In addition, the morbid duration was computed more precisely using the Markov model, which also affected the larger DALY. This study provided several strategies for reducing the disease burden of T2DM. Primary prevention may reduce the YLD of non-complicated T2DM, and secondary prevention such as strict management of blood glucose may contribute to controlling DALYs caused by T2DM complications and YLL by any cause. In addition, this new approach suggested a method for integrating specific diseases and their related states for the estimation of DALYs, even in the incidence-based approach.

## Supporting information

**S1 Fig. Estimated DALYs by gender and age group.** DALYs, disability-adjusted life years.
(TIF)

**S2 Fig. Estimated YLLs by gender and age group.** YLLs, years of life lost.
(TIF)

**S3 Fig. Estimated YLDs depending on complications.** YLDs, years lived with disability.
(TIF)

**S1 Table. Ingredient code of anti-diabetic drugs.**
(DOCX)

**S2 Table. ICD-10 codes of diabetic complications.** ICD, International Classification of Disease.
(DOCX)

## Acknowledgments

We would like to thank the National Health Insurance Service of Korea for providing database for our research. We also thanks Dr. Woo Je Lee for his thoughtful advice on clinical aspects.

## Author Contributions

**Conceptualization:** Seok-Jun Yoon, Min-Woo Jo.

**Data curation:** Juyoung Kim.

**Formal analysis:** Juyoung Kim.

**Funding acquisition:** Seok-Jun Yoon.

**Methodology:** Juyoung Kim, Seok-Jun Yoon, Min-Woo Jo.

**Project administration:** Juyoung Kim.

**Software:** Juyoung Kim.

**Validation:** Juyoung Kim, Min-Woo Jo.

**Visualization:** Juyoung Kim.

**Writing – original draft:** Juyoung Kim, Min-Woo Jo.

**Writing – review & editing:** Juyoung Kim, Seok-Jun Yoon, Min-Woo Jo.

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
