## [Decision Letter · Decision Letter 0]

15 Oct 2020

PONE-D-20-28110

Estimating the disease burden of Korean type 2 diabetes mellitus patients considering its complications

PLOS ONE

Dear Dr. Jo,

Thank you for submitting your manuscript to PLOS ONE. After careful consideration, we feel that it has merit but does not fully meet PLOS ONE’s publication criteria as it currently stands. Therefore, we invite you to submit a revised version of the manuscript that addresses the points raised during the review process.

A **rebuttal letter** that responds to **EACH** point raised by the academic editor and reviewer(s). You should upload this letter as a separate file labeled 'Response to Reviewers'.A **marked-up copy** of your manuscript that highlights changes made to the original version. You should upload this as a separate file labeled 'Revised Manuscript with Track Changes'.An **unmarked version** of your revised paper without tracked changes. You should upload this as a separate file labeled 'Manuscript'.

We look forward to receiving your revised manuscript.

Kind regards,

Brecht Devleesschauwer

Academic Editor

PLOS ONE

Journal Requirements:

2. Please upload a copy of Figure 4, to which you refer in your text on page xx. If the figure is no longer to be included as part of the submission please remove all reference to it within the text.

Additional Editor Comments:

All reviewers have raised some major issues regarding the current version of the manuscript. In particular, please make sure that all methodological choices are described in a transparent way. Also note that PLOS ONE does not provide copyediting for submitted manuscripts and that the submission of a manuscript in clear and correct English is one of the criteria for publication at PLOS ONE (http://journals.plos.org/plosone/s/criteria-for-publication#loc-5). We therefore advise you to seek help from a colleague with a high level of English proficiency or to make use of a professional copyediting service.

Please also note that PLOS ONE evaluates research on scientific and methodological validity, and not on perceived significance.

In your revision note, please include EACH of the reviewer comments, provide your reply, and when relevant, include the modified/new text (or motivate why you decided not to modify the text). Note that failure to do so may result in a rejection of the manuscript.

Reviewers' comments:

Reviewer's Responses to Questions

**Comments to the Author**

1. Is the manuscript technically sound, and do the data support the conclusions?

Reviewer #1: Partly

Reviewer #2: Yes

Reviewer #3: Partly

2. Has the statistical analysis been performed appropriately and rigorously? 

Reviewer #1: I Don't Know

Reviewer #2: Yes

Reviewer #3: N/A

3. Have the authors made all data underlying the findings in their manuscript fully available?

Reviewer #1: No

Reviewer #2: Yes

Reviewer #3: No

4. Is the manuscript presented in an intelligible fashion and written in standard English?

Reviewer #1: Yes

Reviewer #2: No

Reviewer #3: No

5. Review Comments to the Author

Reviewer #1: While national burden of disease studies are of value, since they are based on known data from national data sources, and may be more comfortable to national poliy makers, they are not comparable to the results produced in the global GBD study. Thus publishing national studies in international journals is hardly meaningful. It could have been interesting to compare findings from this study with findings using the data on South Korea from the international burden of disease database, but this is not done. Furthermore, the specific aims of this study, even for national purposes, is not clear. There is no comparison to other countries, there is no comparison to other diseases, there is no data on risk factors (that are available in the GBD study), So the interest of these findings for policy making is not clear.

While the data sources used are appropriate, and the application of sequalae of Diabetes type 2 is appropriate, and in line with the GBD study, I can not judge the Markov methodology to arrive at the overall burden of disease. While probably statistically well done, the authors do not give a convincing argument for the use of this methodology. I can only note that while the prevalence of diabetes type 2 seems quite in line with international studies, the number of DALYs/100000 is much higher than those obtained in the GBD study. See e.g. Khan, Hashim et al Int J Epidemiology and Global Health Volume 10, Issue 1, March 2020, Pages 107 – 11. This and other publications in the global network assessing burden of disease due to diabetes type 2 should be referenced.

Reviewer #2: The article presents an interesting extension to the general and well-spread methodology that is normally used in burden of disease studies.

The article though has some language inaccuracies (multiple grammar/spelling errors) that compromise readability. I suggest a professional revision of the language that would also help to avoid colloquialisms.

Methods section:

In general, there are few methodological choices that are lacking an explanation.

• At pag 4 when you say: “Initial states were assumed to stay in their states for a defined single cycle, then move to other Markov states, including themselves.” What do you mean with “including themselves”? That they can stay in the same health state? Please rephrase

• Why use an age limit of 30 years for identification of T2DM cases

• Could you please provide a better explanation of your approach for the incident and prevalent cases on pag 5

• Could you maybe describe what do you mean for washout period for the case definition? and why do you use it

• Are the causes of deaths also retrieved from the NHIS?

• Could you please describe better the process used for the sensitivity analysis? Why did you choose specifically that subcategories? What was the process? I don’t understand the rationale in modifying the disease duration just for that subcategory.

Results section:

• Could you please remind the year of reference when you describe the results? Is 2016 for both incidence and prevalence?

• “While there was an increasing tendency in the incidence rates with age, the age specific diabetes incidence rate exceeded the aggregate incidence rate (12.5 per 1,000 people) from the 50–54 age group in men.” I would change the first part of the sentence. In the way it is written it seems like you are describing 2 opposite trends.

• Pag 9 – “The occurrence probabilities of complications in the incident cases of diabetes were higher than in the existing prevalent cases of diabetes in relatively young age groups in both genders.” This is interesting but I would think about another way of writing the concept. I don’t see the reversed tendency expressed in the sentence that is following.

• I would not use the term “mountain shapes”. Please switch to bell shape or normally distributed.

Reviewer #3: General comments:

The paper aims at estimating the burden of disease due to diabetes. It builds on previous Korean study where no severity was taken into account. The study takes into account diabetes complications and uses incidence approach with a help of a multi-state Markov model.

The article will improve enormously if the methodology is clearer described. The way the article is written, it is understandable only for insiders and people with deep knowledge in the topic. In order to be understood from a broader audience, some of the terminology should be further explained. For instance, in simpler language the Markov model should be explained where the terminology should be clearer: Markov states, cycles, discount rates, etc.

Furthermore, many assumptions in the model (pages 6 and 7) stay unclear why they are taken. Better descriptions and arguments must be brought up.

There is a mismatch between the numbering of the graphs. The supporting files have other numbers than the one in the text.

In my opinion, the article also needs English proof. Some of the sentences are not really clear.

Further suggestions:

Page 3, Introduction, first sentence: “diabetes is leading cause, accounting …” –> is this globally meant or for Korea?

Second sentence: (…) jumped from 18th to 11th between (…) -> not clear what this is. Is this place of ranking?

Third sentence and fourth sentence have a repetition: “Considering substantial prevalence of diabetes, the diabetes burden is expected to be greater in the future” and then “(…) the burden of diabetes is expected to increase, giving increasing prevalence.”

Fifth sentence: “(…) T2DM dominates the vast majority so it accounts for (…)” -> the vast majority of what?

Page 3, Paragraph 2: here it is mentioned DisMod II for the first time. Many readers would not know what it is, who has developed it, how it functions. A conventional tool is not really the right description; the access to it is very limited.

Disease period -> is here duration meant? And why it should be important for diabetes, when this disease is irreversbale?

Page 4, paragraph 1: As mentioned above, several terminologies should be better explained.

Sentence: “Initial states were assumed (…), including themselves. -> please edit, it is not clear what is meant.

Sentence after: why are diabetes and diabetic states assumed to be irreversible? -> they are irreversible, there is no place for assumptions here.

Sentence next: why is the assumption for five cycles? What is the justification?

Sentence next: “Among the 12 conceptual transitions (…)” -> which are these 12 transitions?

Page 6, paragraph 1: Here should be explained what a washout period is and why a three-year washout period was assumed? What is the justification for this?

Page 6, formula: Is time here meant to be duration?

Page 6, last sentence: should be explained what smoothing methods imply and why they are used.

Page 7, paragraph 1: Not clear why there was a replacement in case a mortality probability was less than the general mortality probability. And why in age group 30-34 zero mortalities were replaced with general death probabilities?

Page 7, paragraph 2: “(…) by the duration between transitions of interest” -> what are the transitions of interest here?

Page 7, last paragraph: “A sensitivity analysis was conducted for the reverse transitions (…)” -> what are here reverse transitions in case of diabetes?

Page 8, paragraph 1, last sentence: University name not readable.

Page 12, Tables 2A and 2B: In the methods section it is written that sensitivity analysis is performed for ages 60 +. Why these tables include information on younger age groups? There are also no differences in the numbers there.

Page 13, paragraph 1: “The disease burden (…) than that in the previous KNBD study -> please include a reference here, even of it is repeating.

Page 15, last sentence: “three years of experience” -> I don’t think “experience” here is the right word.

Figures 2 and 3 in the text (in the files they have the names Figure 1 and 2): These figures look like age pyramids and intuitively one expects the youngest age groups to start from the bottom and go up. Consider to change it.

Files Fig4.tif and S Fig3.tif: the figures have the same labels but present different results.

6. PLOS authors have the option to publish the peer review history of their article (what does this mean?). If published, this will include your full peer review and any attached files.

Reviewer #1: **Yes: **Peter Allebeck

Reviewer #2: **Yes: **Vanessa Gorasso

Reviewer #3: No

---

## [Author Response · Author response to Decision Letter 0]

12 Dec 2020

Our manuscript has been proofread by Editage (www.editage.co.kr). Responses to the reviewers' comments are presented in the Letter of Response to Reviewers.

---

## [Decision Letter · Decision Letter 1]

25 Jan 2021

Estimating the disease burden of Korean type 2 diabetes mellitus patients considering its complications

PONE-D-20-28110R1

Dear Dr. Jo,

We’re pleased to inform you that your manuscript has been judged scientifically suitable for publication and will be formally accepted for publication once it meets all outstanding technical requirements.

Kind regards,

Brecht Devleesschauwer

Academic Editor

PLOS ONE

Additional Editor Comments (optional):

Reviewers' comments:

Reviewer's Responses to Questions

**Comments to the Author**

1. If the authors have adequately addressed your comments raised in a previous round of review and you feel that this manuscript is now acceptable for publication, you may indicate that here to bypass the “Comments to the Author” section, enter your conflict of interest statement in the “Confidential to Editor” section, and submit your "Accept" recommendation.

Reviewer #2: All comments have been addressed

2. Is the manuscript technically sound, and do the data support the conclusions?

Reviewer #2: Yes

3. Has the statistical analysis been performed appropriately and rigorously? 

Reviewer #2: Yes

4. Have the authors made all data underlying the findings in their manuscript fully available?

Reviewer #2: Yes

5. Is the manuscript presented in an intelligible fashion and written in standard English?

Reviewer #2: Yes

6. Review Comments to the Author

Reviewer #2: The methods section has been further expanded and the methodological choice are clearer now.

The manuscript was also revised in terms of English language and it looks more fluent.

7. PLOS authors have the option to publish the peer review history of their article (what does this mean?). If published, this will include your full peer review and any attached files.

Reviewer #2: **Yes: **Vanessa Gorasso

---

## [Editor Report · Acceptance letter]

27 Jan 2021

PONE-D-20-28110R1 

Estimating the disease burden of Korean type 2 diabetes mellitus patients considering its complications 

Dear Dr. Jo:

I'm pleased to inform you that your manuscript has been deemed suitable for publication in PLOS ONE. Congratulations! Your manuscript is now with our production department. 

Kind regards, 

on behalf of

Prof. Dr. Brecht Devleesschauwer 

Academic Editor

PLOS ONE